# Recognition of Postoperative Cystography Features by Artificial Intelligence to Predict Recovery from Postprostatectomy Urinary Incontinence: A Rapid and Easy Way to Predict Functional Outcome

**DOI:** 10.3390/jpm13010126

**Published:** 2023-01-08

**Authors:** I-Hung Shao, Hung-Cheng Kan, Hung-Yi Chen, Ying-Hsu Chang, Liang-Kang Huang, Yuan-Cheng Chu, Po-Hung Lin, Kai-Jie Yu, Cheng-Keng Chuang, See-Tong Pang, Chun-Te Wu

**Affiliations:** 1Division of Urology, Department of Surgery, Linkou Chang Gung Memorial Hospital, Taoyuan 333005, Taiwan; 2College of Medicine, Chang Gung University, Taoyuan 333323, Taiwan; 3Graduate Institute of Clinical Medical Sciences, College of Medicine, Chang Gung University, Taoyuan 333323, Taiwan; 4Department of Urology, Chang Gung Memorial Hospital at Keelung, Keelung 204201, Taiwan; 5Division of Urology, Department of Surgery, New Taipei Municipal TuCheng Hospital, Chang Gung Memorial Hospital, New Taipei 236017, Taiwan

**Keywords:** radical prostatectomy, urinary incontinence, cystography, deep learning, artificial intelligence, postprostatectomy incontinence

## Abstract

Purpose: Post-operative cystography has been used to predict the recovery of postprostatectomy urinary incontinence (PPI) in patients with localized prostate cancer. This study aimed to validate the predictive value of cystography for PPI and utilize a deep learning model to identify favorable and unfavorable features. Methods: Medical records and cystography images of patients who underwent robotic-assisted radical prostatectomy for localized prostate cancer were retrospectively reviewed. Specific cystography features, including anastomosis leakage, a downward bladder neck (BN), and the bladder neck angle, were analyzed for the prediction of PPI recovery. Favorable and unfavorable patterns were categorized based on the three cystography features. The deep learning model used for transfer learning was ResNet 50 and weights were trained on ImageNet. We used 5-fold cross-validation to reduce bias. After each fold, we used a test set to confirm the model’s performance. Result: A total of 170 consecutive patients were included; 31.2% experienced immediate urinary continence after surgery, while 93.5% achieved a pad-free status and 6.5% were still incontinent in the 24 weeks after surgery. We divided patients into a fast recovery group (≤4 weeks) and a slow recovery group (>4 weeks). Compared with the slow recovery group, the fast recovery group had a significantly lower anastomosis leakage rate, less of a downward bladder neck, and a larger bladder neck angle. Test data used to evaluate the model’s performance demonstrated an average 5-fold accuracy, sensitivity, and specificity of 93.75%, 87.5%, and 100%, respectively. Conclusions: Postoperative cystography features can predict PPI recovery in patients with localized prostate cancer. A deep-learning model can facilitate the identification process. Further validation and exploration are required for the future development of artificial intelligence (AI) in this field.

## 1. Introduction

Worldwide, prostate cancer is the second most common malignancy, and the fifth most lethal cancer in men [1]. Treatments, including radical prostatectomy (RP) and radiation therapy, such as external beam radiotherapy or brachytherapy, are recommended for localized prostate cancer in therapy guidelines [2]. Among local therapies, RP has high efficacy for the treatment of localized prostate cancer, with a 97% 5-year survival rate after radical surgery [3]; with good oncological outcomes, and potential side effects are not inevitable, including urinary incontinence, and sexual or bowel dysfunction [4], which would have a negative impact on quality and eventually medical and economic burden to both patients and healthcare systems [5]. 

As one of the most bothersome functional complications, postprostatectomy urinary incontinence (PPI) affects up to 20% of patients after RP [6]. The definition of PPI varies among different studies. Generally, the PPI rate according to “no pad use” at 12 months ranged from 4% to 31% in previous research [7]. The occurrence of PPI is multi-factorial and can be related to preoperative, intraoperative, or postoperative factors, either from patients themselves or surgeons [8]. 

The stigmatization of PPI affects patients, causing depression and anxiety, and also substantially reduces their quality of life [9]. In addition to the impact on patients, PPI results in a marked medical cost burden, which was estimated to be between $19 and $32 billion in the USA [10]. In patients with localized prostate cancer who received a RP, it was estimated that 3% of patients underwent at least one incontinence procedure during a median follow-up of 5 years. Among the treatment modalities for PPI, male slings were the most common incontinence procedure followed by an artificial urinary sphincter (AUS), which was the most expensive option [11]. 

Previous studies have suggested several risk factors for PPI occurrence; pre-operative factors, such as the patient age, body mass index (BMI), history of transurethral resection of the prostate, prostate volume, and previous lower urinary tract symptoms were reported. In addition, prostate cancer characteristics, including the clinical stage, and the prostate-specific antigen (PSA) level, are correlated with PPI [12]. Intraoperative factors including the surgical method, preservation of the neurovascular bundle, preservation of the bladder neck, and vesicoureteral anastomosis, were also well established to correlate with incontinence recovery [13]. Factors, such as older age, diabetes mellitus, prior radiation therapy, and a history of neurological comorbidities, had also been reported to be associated with an increased likelihood of incontinence after surgery [11]. 

To predict the occurrence of PPI more precisely, multiple studies have utilized different image-based tools to predict PPI and attained acceptable outcomes [14]. We previously reported that a specific pattern of postoperative cystography could predict PPI. Specific features included a downward bladder neck, the bladder neck angle, and anastomosis leakage [15]. The predictive value of these specific features in cystography for PPI is now well established [16,17,18,19,20,21,22,23,24]. 

Medical image is a tool for disease diagnosis and the prediction of the disease prognosis. Artificial intelligence (AI) has been widely used for medical image analysis in recent years. Deep learning is a state-of-the-art tool for the medical image analysis approach; it also brings revolutionary changes in lesion detection and classification. Applying a deep-learning-based medical images analysis could provide decision support to clinicians and reduce daily working loading. In this study, we aimed to validate the predictive value of cystographic features for PPI. We also applied a deep learning model to recognize cystography after RP, which can distinguish between favorable and unfavorable patterns for recovery from PPI.

## 2. Materials and Methods

### 2.1. Patient Selection and PPI Evaluation

We retrospectively collected data from patients who underwent robot-assisted radical prostatectomy (RARP) performed by a single surgeon between May 2015 and January 2019. The study was approved by the institutional review board of our hospital (Number 201504523B0C501). The study was conducted in accordance with the ethical principles of the Declaration of Helsinki (2013). The requirement for informed consent was waived by the IRB due to the retrospective design of the study.

Clinical information, including age, BMI (body mass index), the initial prostate specific antigen (PSA), and the prostate volume, was obtained via medical chart reviewing. Intra-operative factors, such as the operative time and blood loss, were also recorded. During the surgery, neurovascular bundle (NVB) preservation was decided and performed according to the tumor location and invasion on the pre-operative image and intraoperative judgement by the surgeon. Patients with any existing urinary incontinence (including urge incontinence, stress incontinence, or mixed incontinence) before RARP were excluded in this study.

After the RARP, the urinary catheters were routinely removed one week after surgery if no anastomosis leakage was found on cystography. However, if the minor anastomosis leakage was noted, the urinary catheter would be kept for one more week. In patients with severe anastomosis leakage, the urinary catheter would be removed based on the clinical judgement of the physician.

The continence status was evaluated immediately after the catheter removal and at 4, 12, and 24 weeks after surgery, followed by 12 -weeks intervals by the same physician, based on the data entered in a 3-day voiding diary and the patient report at the outpatient department. No patients were lost to follow-up during the evaluation period.

### 2.2. Cystography Pattern

All patients were followed up for at least 12 months after the RARP. Postoperative cystography was performed within 1 week of the RARP before the urethral catheter removal. Urinary bladder patterns on cystography were analyzed according to four categories: urine leakage on cystography, length of the downward bladder neck, the bladder shape, and the bladder neck angle. 

Urine leakage on cystography was indicated by contrast medium extravasation at the urethra-bladder neck junction (Figure 1A). The length of the downward bladder neck was defined as the length of the bladder neck below the lower margin of the pelvic inlet (Figure 1B). The bladder shape was described by the bladder height, bladder width, and height/width ratio. The bladder neck angle was measured as the angle between the bladder neck and the bilateral bladder margins over the pelvic inlet (Figure 1C).

All cystography images were analyzed by a single radiologist who was blinded to the patient PPI status.

### 2.3. Artificial Intelligence Deep Learning Training

Criteria for cystography pattern categorization

We defined an unfavorable cystography pattern as a downward bladder neck, sharp bladder neck angle, and anastomosis leakage on cystography. The criteria for unfavorable and favorable groups for training the deep learning model were defined with a downward BN < or ≥ 0.8 cm (median value in our study), BN angle < or ≥ 115 degrees (median value in our study), and anastomosis leakage as below:

Favorable: downward BN < 0.8 cm, or BN angle ≥ 115°, without anastomosis leakage.

Unfavorable: downward BN ≧ 0.8 cm, and BN < 115, or anastomosis leakage.

### 2.4. Image Preprocessing

All cystography images were cropped to create region-of-interest (ROI) images (Figure 1D). The bilateral margin of the ROI images is the bilateral wall of the bladder, as shown in the cystography; the upper margin is the ischial spine and the lower margin is the inferior pubic ramus. Within this defined area, we adjusted the upper and lower margins of the cystography images to fit the horizontal part of the pubic bone in the middle third of the ROI images. The ROI images were manually indicated by two experienced urologists on each cystography image. If the cystography image was labeled, such as numbers of contrast amounts, arrows to indicate the contrast leakage site, or with imaging artifacts, such as metallic material in our ROI, it was abandoned. 

### 2.5. Datasets Setting

Patients were divided into favorable (n = 80) and unfavorable (n = 77) groups. We split 10% of all images as test data (n = 16, favorable group n = 8, and unfavorable group n = 8). For the training data (n = 141, favorable group n = 72, and unfavorable group n = 69), 90% of all images were divided into 80% and 20% for a 5-fold cross-validation. 

### 2.6. Establishment of Model and Training Details

The deep learning model used for transfer learning was ResNet 50 and the weights were trained on ImageNet. The model parameter settings were epochs = 30, learning rate = 10^−5^, and batch size = 8. We used 5-fold cross-validation to reduce bias. After each fold, we used test data to evaluate the model performance. 

### 2.7. Statistical Analysis

All cystography parameters were evaluated for their relationship with the PPI status. Variables with a 20% significance level in the univariate analysis were followed by a multiple logistic model, which was visualized by Nomogram using R packages. A statistical analysis was performed using IBM SPSS, version 19 (IBM Corp, Armonk, NY, USA), for independent *t*-tests, the chi-square test, Kaplan Meier’s survival analysis, and the receiver operating characteristic curve analysis. The significance level was set at 0.05. The sensitivity and specificity of the machine learning model for classification were calculated using a fixed cut-off of 0.5.

## 3. Results

A total of 170 consecutive patients were included in this study. The general characteristics of the included patients are summarized in Table 1. The mean age was 66.9 years and the mean initial serum PSA level was 17.3 ng/ml. The high-risk group, based on D’Amico criteria, accounted for the majority of the patients in this study (75.9%). Most patients had a pathological T2 stage disease (69.4%), and 63% had a Gleason score of 7 (4 + 3 or 3 + 4). A bilateral NVB preservation was received by 54.7% of patients, while no NVB was preserved in 28.8% of patients. Of the patients, 31.2% experienced immediate urinary continence after surgery, while 93.5% achieved a pad-free status and 6.5% were still incontinent 24 weeks after surgery. 

We divided the patients into fast recovery (≤4 weeks) and slow recovery groups (>4 weeks). The differences in cystography patterns between these two groups were compared. In the univariate analysis, the patients who recovered from PPI within 4 weeks after surgery had significantly lower anastomosis leakage, less of a downward bladder neck, and a larger bladder neck angle than those who recovered from PPI more than 4 weeks after surgery. Detailed results are listed in Table 2.

We used a receiver-operating characteristics (ROC) curve analysis to evaluate the predictability of the downward bladder neck and bladder neck angle for a pad-free status 4 weeks after surgery. The area under the curve (AUC) was 0.777 and 0.745 for the downward bladder neck and bladder neck angle, respectively (Figure 2A).

Based on the above results, we selected anastomosis leakage, a downward BN, and the BN angle on cystography as unfavorable cystography features for PPI recovery. The model is visualized in Figure 3 with its logistic model equation: logit(p)=−1.1163+−0.8843×(anastomosis leak=Yes)−1.2889×(Downward BN)+0.0205×BN angle, where *p* is the probability of being pad free in 4 weeks. Further, we categorized cystography images with the downward BN, acute BN angle, and anastomosis leakage as the unfavorable group, while those with less of a downward BN, an obtuse BN angle, and without anastomosis leakage were categorized as the favorable group. 

The test data used to evaluate the model performance demonstrated that the average of the 5-fold cross validation accuracy, sensitivity, and specificity were 92.5%, 87.5%, and 97.5%, respectively. Across the 5-fold cross-validation scheme, the best accuracy, sensitivity, and specificity were 93.75%, 87.5%, and 100%. Figure 2B shows the ROC curves for the best performing model. The results of the test data used to evaluate the trained model by the 5-fold cross validation scheme are summarized in Table 3. We used Gradient-weighted Class Activation Mapping (Grad-CAM) to determine the most important region for making decisions using our model (Figure 4).

## 4. Discussion

Based on previous literature, among the image-based predictors for PPI, including a intravesical protrusion, the membranous urethral length (MUL), the prostatic volume, and periurethral fibrosis [25,26,27,28], MUL on the preoperative MRI is one of the most discussed [14]. More than 20 articles have discussed the predictive value of MUL for PPI, with a significant correlation clearly identified between a longer length pre-surgery and continence after surgery, suggesting that MUL could be an independent predictor for early recovery from PPI [29]. 

Although the static preoperative measurement of MUL by MRI has shown promise, preoperative factors cannot adequately represent the postoperative status. For example, a patient with an ideal MUL before RP may sacrifice MUL during surgery due to cancer control, peri-prostate adhesion, or the surgeon’s technique. To evaluate PPI recovery more precisely, postoperative imaging is clearly a better choice.

Cystography is a low-cost and convenient exam with minimal adverse effects, which had been used in patients with suspicious anastomosis leakage after RP. Jeong et al. first reported that the location of vesicourethral anastomosis on postoperative cystography could predict early recovery from PPI in 2011 [30]. Compared to preoperative imaging, postoperative imaging is theoretically more accurate in predicting postoperative bladder and sphincter function. We further reported that, not only the bladder neck position, but also a specific cystography pattern combining the downward bladder neck, bladder neck angle, and anastomosis leakage could predict PPI [15]. 

Following our pilot study, other studies have confirmed the predictive value of the three specific cystography features for PPI mentioned in our study: more of a downward bladder neck level [16,19,20,22,24], a sharp bladder neck angle [18], and anastomosis leakage [21] are associated with a delayed and lower probability of recovery from PPI.

In this study, we validated the value of postoperative cystography patterns in predicting PPI. All three adverse features in cystography were significant predictors of PPI and had a high AUC of ROC. We then further divided the cystography pattern into favorable and unfavorable for recovery from PPI based on the downward bladder neck, bladder neck angle, and anastomosis leakage. We established an AI model that can automatically recognize favorable or unfavorable cystographic features for PPI.

Convolutional neural networks (CNNs) have been widely used in medical image classification and have demonstrated remarkable performance in many medical fields. Esteva et al. [31] demonstrated that CNNs are capable of classifying skin cancers with a level of competence comparable to that of dermatologists. Arevalo et al. [32] used CNNs to classify mass lesions in mammography film images and achieved good results as a representation from 79.9% to 86.0% in terms of the area AUC of the ROC curve. Khondker et al. [33] found that vesicoureteral reflux (VUR) grades can be differentiated using a classifier and achieved an accuracy of 0.84 on external validation. However, the ability of CNNs to classify different bladder shapes from cystography images had not yet been fully studied.

A CNNs deep learning model was used in our study to differentiate favorable and unfavorable shapes of bladders from cystography, and achieved the highest accuracy of 0.94, sensitivity of 0.88, and specificity of 1 (Table 3). As expected, Grad-CAM showed that the important regions for making decisions by our model are around the bladder neck, which is compatible with our criteria for demonstrating favorable and unfavorable patterns of the bladder neck for recovery from PPI.

This is the first study to utilize a deep learning model to categorize postoperative cystography images into favorable and unfavorable patterns, which have been demonstrated to be predictive of recovery from PPI. Compared with preoperative images, postoperative images can provide more accurate and representative information of the same individual after surgery. With the assistance of a deep learning model, we can more easily and precisely predict recovery from PPI.

A combination of AI and radiomics has been increasingly utilized in the field of urology, including prostate cancer in recent years. In addition to our study, Gravina et al. also demonstrated that combining clinical information and radiological parameters with machine learning can better detect the prostate malignancy probability of equivocal prostate lesions [34]. 

The key factors to improve the accuracy of the deep learning model for image identification are the quality and case number of images we provided. To improve the quality and make the image consistent, we cropped the cystography images manually to standardize the border of the image and exclude unnecessary or irrelevant information on the cystography image. Unlike some diseases with a high prevalence which could generate numerous images for training, such as a chest X-ray for pneumonia, the number of patients with prostate cancer who received RARP is relatively smaller. To solve this problem, we applied the transfer leaning approach, which has been widely applied in the AI field since Pratt introduced it in 1992 [35]. While the training data is small, transfer learning is a common method to develop a deep learning model. In transfer learning, the deep learning model has been well-trained with a large data set, and the weights that the model built from the large data set could be transferred to a separate task. By using the transfer learning approach, we can complete our specific task with a relatively small data set and achieve adequate accuracy.

This study had some limitations. First, the retrospective study design and relatively small sample size may require a further larger prospective cohort to confirm. Second, the quality of the cystography images may influence the identification of specific features. For example, in our study, we excluded images with confounding factors, such as letters marking the infusion water volume around the targeted area. Third, although we used independent test data to evaluate the model’s performance, all of the data was from a single institution. If we could use internal and external multi-institutional data to assess our model, the result would be more confident. As a result, the performance of a deep learning model can achieve the needs in the clinical practice and remains consistent over time. Different cystography protocols may also impact the identification accuracy of the AI system. In addition, PPI is a multifactorial complication that may be caused by the patient preoperative condition or perioperative factors. Postoperative cystography features can only represent partial aspects of PPI. Further efforts are needed to integrate cystography features with previously well-established clinical predictors, such as the patient age, BMI, urethral length spared, quality of anastomosis, preservation of bladder neck, and rehabilitation programs. The combination of pre-operative, intra-operative, and post-operative factors can definitely better predict the occurrence and recovery of PPI. Furthermore, the prediction system may be modified and applied to predict the response of sling surgery for PPI. To date, the application of the AI system in cystography classification is still in a preliminary stage, and much remains to be explored. With a sufficient data set and computational techniques, developing a robust deep learning model for cystography classification could be expected in the future.

## 5. Conclusions

Postoperative cystography features, including the downward bladder neck, bladder neck angle, and anastomosis leakage, can predict recovery from PPI in patients with localized prostate cancer after RARP. To facilitate the identification process, we developed an AI deep learning system to categorize cystography images as a favorable pattern and an unfavorable pattern for PPI recovery. This is the first time that AI has been applied in an image-based prediction for PPI. Further validation and combination with clinical information are required for the future development of AI to achieve a better prediction value.

## Figures and Tables

**Figure 1 jpm-13-00126-f001:**
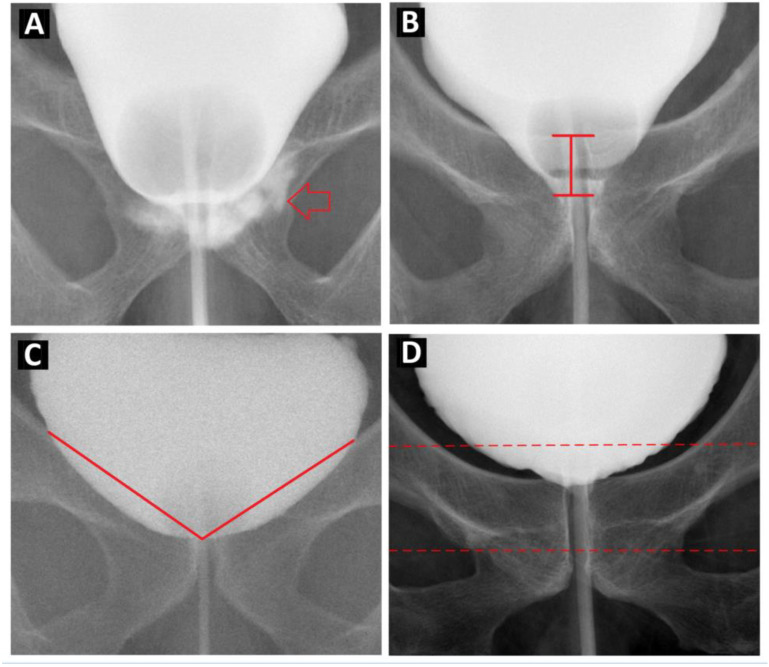
Illustration of Cystography feature. (**A**). Urinary leakage in cystography was defined as contrast leakage at bladder neck and urethra anastomosis site (red arrow). (**B**). The distance of downward bladder neck was defined as the length of bladder neck below the pelvic inlet lower margin, shown as the length of the red arrow between the two black horizontal lines. (**C**). The bladder neck angle was measured as the angle of bladder neck to bilateral bladder margin over pelvic inlet. The angle was shown as the angle between the two red lines. (**D**). The bilateral margin of the ROI images is the bilateral wall of the bladder, as shown in the cystography; the upper margin is the ischial spine and the lower margin is the inferior pubic ramus. Within this defined area, we adjusted the upper and lower margins of the cystography images to fit the horizontal part of the pubic bone in the middle third of the ROI images.

**Figure 2 jpm-13-00126-f002:**
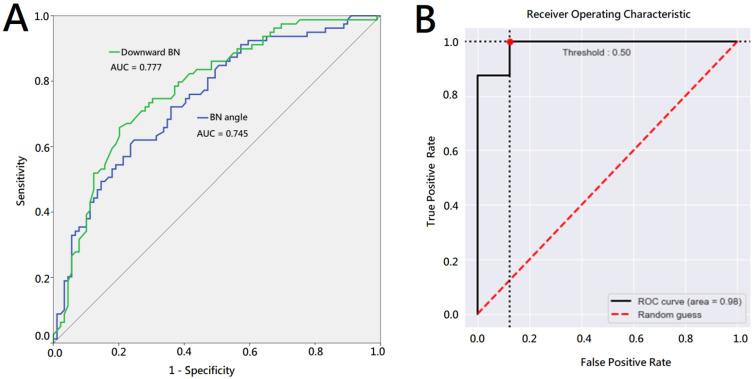
Receiver-operating characteristics curve analysis. (**A**) ROC curve of downward bladder neck and bladder neck angle for pad-free status 4 weeks after surgery. (**B**) ROC curves for the best performing deep learning model to predict unfavorable and favorable cystography group.

**Figure 3 jpm-13-00126-f003:**
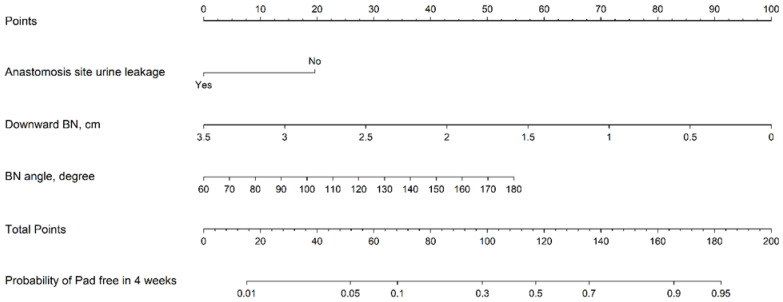
Nomogram of cystography features to predict pad free probability in 4 weeks.

**Figure 4 jpm-13-00126-f004:**
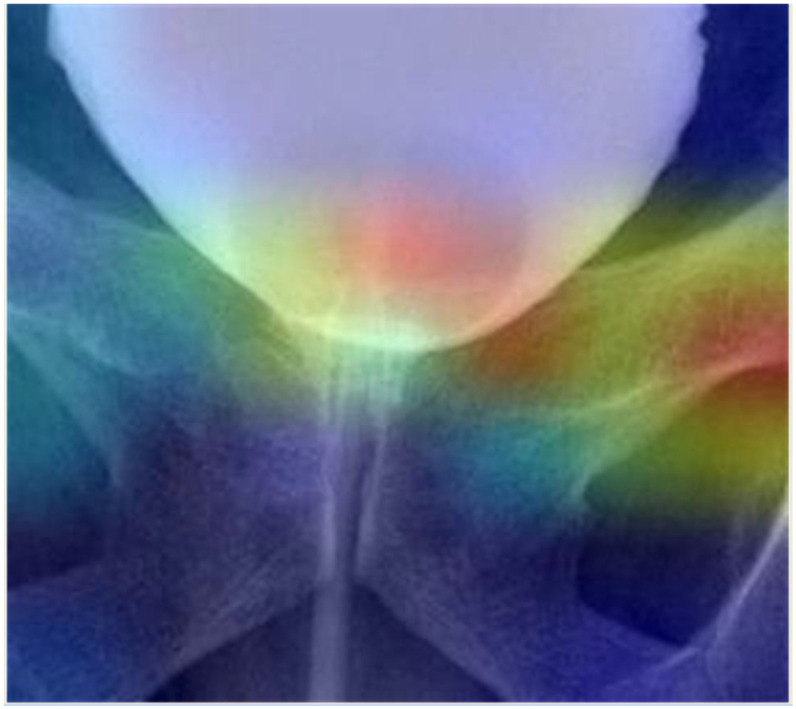
Gradient-weighted Class Activation Mapping (Grad-CAM). The most important region for making decisions using our model, as determined by the Grad-CAM.

**Table 1 jpm-13-00126-t001:** Clinicopathological patient characteristics.

Age, Years (SD, Range)		66.9 (6.1, 51–80)
BMI, kg/m^2^ (SD, range)		25.2 (3.2, 15.9–36)
Initial PSA, ng/ml (SD, range)		17.3 (18.3, 2.29–150)
Prostate volume, gm (SD, range)		40.6 (17.2, 12–116)
D’Amico risk group (%)	Low	12 (7.1)
	Intermediate	29 (17.1)
	High	129 (75.9)
Operative time, mins (SD, range)		275.1 (49.4, 152–481)
EBL, ml (SD, range)		108.1 (83.3, 20–550)
Pathological stage (%)	pT2	118 (69.4)
	pT3	51 (30.0)
	pT4	1 (0.6)
Surgical margin (%)	Negative	116 (68.2)
	Positive	54 (31.8)
NVB preservation (%)	Monolateral	28 (16.5)
	Bilateral No	93 (54.7) 49 (28.8)

Bladder neck reconstruction (%)	Yes	81 (47.7)
	No	89 (52.4)
Continence recovery time (%)	Immediate	53 (31.2)
	4 weeks	81 (47.6)
	12 weeks	144 (84.7)
	24 weeks	159 (93.5)

BMI = body mass index; PSA = prostate specific antigen; EBL = estimated blood loss; NVB = neurovascular bundles; PPI = post-prostatectomy incontinence; SD = standard deviation.

**Table 2 jpm-13-00126-t002:** Cystography patterns in fast & slow recovery groups.

	Pad Free ≦ 4 Weeks	Not Pad Free in 4 Weeks	Univariate Analysis	Multivariate Analysis
	(N = 81)	(N = 89)	*p* Value	
Anastomosis site urine leakage (%)			* 0.011	* 0.04
Yes	12 (14.8)	29 (32.6)		
No	69 (85.2)	60 (67.4)		
Bladder height, cm (SD, range)	8.04 (1.33, 4.6–11.9)	8.18 (1.37, 5.3–11.4)	0.501	
Bladder width, cm (SD, range)	7.21 (1.16, 4.7–11.0)	7.14 (1.18, 4.3–10.6)	0.677	
Height to width ratio (SD, range)	1.13 (0.19, 0.8–1.7)	1.17 (0.25, 0.6–2.1)	0.210	
Downward BN, cm (SD, range)	0.74 (0.49, 0.1–3.5)	1.31 (0.68, 0.1–3.5)	** <0.001	* 0.011
BN angle, degree (SD, range)	128.37 (18.0, 83.9–170.7)	110.78 (20.2, 62.6–167.2)	** <0.001	0.122

BN = bladder neck; SD = standard deviation; * *p* value < 0.05; ** *p* value < 0.01.

**Table 3 jpm-13-00126-t003:** Accuracy, sensitivity, and specificity of 5-fold machine learning model.

	Accuracy	Sensitivity	Specificity
Fold 1	0.94	0.88	1.00
Fold 2	0.88	0.88	0.88
Fold 3	0.94	0.88	1.00
Fold 4	0.94	0.88	1.00
Fold 5	0.94	0.88	1.00

## Data Availability

The data presented in this study are available on request from the corresponding author. The data are not publicly available due to patient privacy.

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
