# Peer review of "Recognition of Postoperative Cystography Features by Artificial Intelligence to Predict Recovery from Postprostatectomy Urinary Incontinence: A Rapid and Easy Way to Predict Functional Outcome"

_jpm, 2023, doi:10.3390/jpm13010126_

Round 1
Reviewer 1 Report
This is a retrospective study evaluating a postoperative model to predict urinary incontinence. The study is well designed a well written and have promising features.
However, there are some points that I would like to ask the authors.
Minor
Clarify the surgical technique. Lymph node dissection was performed with RARP?. It was shown to be related to urinary incontinence. In addition, the Retzius space was conserved
In all patients the same technique for NVB conservation was used?
Please shorten the title if possible
The sentence …”In patients with localized prostate cancer and received RP…” Change to ”…In patients with localized prostate cancer who received RP…”
Please check the data of the table 1. NVB preservation (%) 28+94+49=171 instead of 170
Table 2. It is not clear the word pad free>4 weeks. Before it says that 93.5% have recovered continence at 24 weeks. So, all patients recovered continence during follow up?
I am not so familiar to this type of model. Is it possible to develop a nomogram based on these results? That way, we can weight the variables and the model can be used by the physicians.
Discussion section please replace radical prostatectomy for RP, as it has been already introduced.
Gravina et al. Please deleted the underliyng
Please limit redundant information.
Author Response
Response to Reviewer 1 Comments
Point 1: Clarify the surgical technique. Lymph node dissection was performed with RARP?. It was shown to be related to urinary incontinence. In addition, the Retzius space was conserved
Response 1: We used MSKCC Nomogram, 5% cut-off, to decide lymph node dissection was performed or not. The Retzius space was not conserved during our procedure.
Point 2: In all patients the same technique for NVB conservation was used?
Response 2: We had specified this part in the materials and methods section as below:
‘’During the surgery, neurovascular bundle (NVB) preservation was decided and performed according to tumor location and invasion on pre-operative image and intraoperative judgement by the surgeon. ”
Point 3: Please shorten the title if possible
The sentence …”In patients with localized prostate cancer and received RP…” Change to ”…In patients with localized prostate cancer who received RP…”
Response 3: Thank you for the suggestion and we had replaced the word “and” with “who” in the manuscript.
Point 4: Please check the data of the table 1. NVB preservation (%) 28+94+49=171 instead of 170
Response 4: Thank you for the precious comment. We had revised the mistake as 93 patients in bilateral preservation group.
Point 5: Table 2. It is not clear the word pad free>4 weeks. Before it says that 93.5% have recovered continence at 24 weeks. So, all patients recovered continence during follow up?
Response 5: Thank you for the precious comment. We had revised the title as “Pad free ≦ 4 weeks” and “Not pad free in 4 weeks”.
Point 6: I am not so familiar to this type of model. Is it possible to develop a nomogram based on these results? That way, we can weight the variables and the model can be used by the physicians.
Response 6: Thanks for the precious comment. We had performed a nomogram analysis based on the three parameters. The nomogram was listed in figure 4.
Point 7: Discussion section please replace radical prostatectomy for RP, as it has been already introduced.
Response 7: Thank you for the advice. We replaced radical prostatectomy for RP as it has been already introduced.
Point8: Gravina et al. Please deleted the underliyng
Please limit redundant information.
Response 8: Thanks for the precious comment. We deleted the underlying and deleted :the sentence as below:
‘’ (including BMI, location of suspicious lesions, serum PSA level, prostate volume, PSA density, and histopathology results) ‘’
Reviewer 2 Report
Thank you for your report, Recognition of Postoperative Cystography Features by Artificial Intelligence to Predict Recovery from Postprostatectomy Urinary Incontinence: A Rapid and Easy Way to Predict Functional Outcome. This is an important addition to the literature.
Suggestions:
Result: A total of 170 consecutive patients were included; 31.2% experienced immediate urinary continence after surgery, while 93.5% achieved a pad-free status 24 weeks after surgery.
CON: this does not add to 100%
Compared with the slow recovery group, the fast recovery group had significantly lower anastomosis leakage rate, less downward bladder neck, and larger bladder neck angle.
Con: can you measure this and give mean and sem.
downward bladder neck, bladder neck angle, and anastomosis leakage.
Con: bladder neck angle to urethra
working loading.
Con work load; more germane is greater accuracy, predict who would benefit from sling
Patients with any existed urinary incontinence
To patients with prior incontinence
After the RARP, the urinary catheters were routinely removed one week; why 7 days others less days
Postoperative cystography was performed within 2 weeks of RARP before urethral catheter removal.
Not clear since you removed the catheter at 1 wk
CON: Rating of downward bladder neck, bladder neck angle, and anastomosis leakage should include numbers. A simple nomogram could be made for predicting incontinence to compare to AI results. That would help to show value of AI.
CON: does downward bladder neck, bladder neck angle, and anastomosis help predict outcomes of sling for incontinence?
Author Response
Response to Reviewer 2 Comments
Point 1: Result: A total of 170 consecutive patients were included; 31.2% experienced immediate urinary continence after surgery, while 93.5% achieved a pad-free status 24 weeks after surgery.
CON: this does not add to 100%
Response 1: Thank you for the precious comment. We revised the sentence as below:
“while 93.5% achieved a pad-free status and 6.5% were still incontinent in 24 weeks after surgery.”
Point 2: Compared with the slow recovery group, the fast recovery group had significantly lower anastomosis leakage rate, less downward bladder neck, and larger bladder neck angle.
Con: can you measure this and give mean and sem.
Response 2: Thank you for the advice. We revised table 2 parameter as mean, SD and range (min-max)
Point 3: downward bladder neck, bladder neck angle, and anastomosis leakage.
Con: bladder neck angle to urethra
Response 3: The bladder neck angle was measured as the angle of bladder neck to bilateral bladder margin over pelvic inlet, as Fig 1C
Point 4: working loading.
Con work load; more germane is greater accuracy, predict who would benefit from sling
Response 4: Thank you for the precious comment. Indeed, in addition to applying cystography pattern to predict PPI, we also want to use the cystography to predict response of male sling for PPI. We are still collecting adequate sample size the generate the conclusion. We will mention this point in the discussion section as below:
“Furthermore, the prediction system may be modified and applied to predict the response of sling surgery for PPI.”
Point 5: Patients with any existed urinary incontinence
To patients with prior incontinence
Response 5: All the patients were totally continent before RARP. We had specified this part in the materials and methods section as below:
“Patients with any existed urinary incontinence before RARP were excluded in this study.”
Point 6: After the RARP, the urinary catheters were routinely removed one week; why 7 days others less days
Postoperative cystography was performed within 2 weeks of RARP before urethral catheter removal.
Not clear since you removed the catheter at 1 wk
Response 6: Thanks for the comment. we had revised the mistake in manuscript as “Postoperative cystography was performed within 1 week of RARP before urethral catheter removal”
Point 7: CON: Rating of downward bladder neck, bladder neck angle, and anastomosis leakage should include numbers. A simple nomogram could be made for predicting incontinence to compare to AI results. That would help to show value of AI.
Response 7: Thanks for the precious comment. We had performed a nomogram analysis based on the three parameters. The nomogram was listed in figure 4.
Point 8: CON: does downward bladder neck, bladder neck angle, and anastomosis help predict outcomes of sling for incontinence?
Response 8: Thank you for the precious comment. Indeed, in addition to applying cystography pattern to predict PPI, we also want to use the cystography to predict response of male sling for PPI. We are still collecting adequate sample size to generate the conclusion. We will mention this point in the discussion section as below:
“Furthermore, the prediction system may be modified and applied to predict the response of sling surgery for PPI.”